# Geoinformatics and Machine Learning for Comprehensive Fire Risk Assessment and Management in Peri-Urban Environments: A Building-Block-Level Approach

Anastasia Yfantidou [1,2,3], Melpomeni Zoka [1], Nikolaos Stathopoulos [1,*], Martha Kokkalidou [1], Stella Girtsou [1], Michail-Christos Tsoutsos [1], Diofantos Hadjimitsis [2,3] and Charalampos Kontoes [1,2]

1 Operational Unit "BEYOND Centre for Earth Observation Research and Satellite Remote Sensing", Institute for Astronomy, Astrophysics, Space Applications and Remote Sensing, National Observatory of Athens, GR-152 36 Athens, Greece; yfantidou@noa.gr (A.Y.); kontoes@noa.gr (C.K.)
2 Department of Civil Engineering and Geomatics, Faculty of Engineering and Technology, Cyprus University of Technology, Saripolou 2-8, Limassol 3036, Cyprus; d.hadjimitsis@cut.ac.cy
3 Eratosthenes Centre of Excellence, Saripolou 2-8, Limassol 3036, Cyprus
* Correspondence: n.stathopoulos@noa.gr

**Abstract:** Forest fires can result in loss of life, damage to infrastructure, and adverse environmental impacts. This study showcases an integrated approach for conducting high-detail fire risk assessment and supporting strategic planning and management of fire events in peri-urban areas that are susceptible to forest fires. The presented methodology encompasses fire hazard modeling, vulnerability and exposure assessment, and in situ observations. Numerous fire hazard scenarios were tested, simulating the spatiotemporal spread of fire events under different wind characteristics. The vulnerability of the studied areas was assessed by combining population data (density and age) and building characteristics, while the exposure parameter employed land value (EUR/m$^2$) as an indicator for qualitatively estimating potential economic effects in the study area. Field campaigns facilitated the identification and recording of critical areas and points, including high-risk buildings and population gathering areas, which subsequently informed the mitigation and fire management planning suggestions. Moreover, field recordings acted as an iterative process for validating and updating the fire risk maps. This research work utilizes state-of-the-art techniques to achieve an analysis of fire risk at a building-block level. Overall, the study presents an applied and end-to-end methodology for effectively addressing forest fire risk in peri-urban areas.

**Keywords:** fire risk assessment; management planning; fire hazard simulations; fire spread; field observations; fire risk prediction; fire vulnerability; GIS

## 1. Introduction

Forest fires are among the most significant natural hazards, with far-reaching consequences for human and animal lives, the environment, properties, the economy, and infrastructure [1]. In recent times, these events have kept rising both in frequency and severity, mainly due to climate change (e.g., higher temperatures, droughts, alteration in precipitation patterns, etc.), rendering fire risk assessment a critical priority [2]. In particular, climate change has increased the vulnerability of forest ecosystems, as it is a major contributor to the rise in forest fires and tree species' inability to adapt to the intensity and frequency of summer droughts [3]. In light of this, peri-urban zones are more prone to wildfires, leaving people's lives, properties, and the natural environment/ecosystem exposed to increased disaster risk. The high fire vulnerability of the peri-urban zones stems mainly from uncontrolled urban sprawl, inadequate urban planning, and lack of appropriate preventive measures [4].

In recent years, Europe has led and supported various critical initiatives to address forest fires, protect ecosystems, and ensure post-fire biodiversity recovery. A significant

milestone in this effort is the formulation of the European strategy on biodiversity (2030) under the European Green Deal, which includes specific actions and measures for fire protection [5]. Furthermore, at a global level, the Sendai Framework for Disaster Risk Reduction 2015–2030 outlines seven clear targets and four priorities for action to prevent new disaster risks and reduce existing ones. These priorities include understanding disaster risk, strengthening disaster risk governance to manage disaster risk, investing in disaster reduction for resilience, and enhancing disaster preparedness for effective response. The Sendai Framework aims to substantially reduce the disaster risk and losses in lives, livelihoods, health, and economic, physical, social, cultural, and environmental assets over the next 15 years [6].

The rising frequency and severity of forest fires and the introduction of novel frameworks and directives for fire risk assessment have sparked numerous studies that aim to enhance our understanding of wildfires and devise effective mitigation strategies. These studies have contributed to advancements in scientific research, data analysis, fire behavior modeling, and prevention strategies. Notably, scientific and technological progress in areas such as remote sensing (RS), geographical information systems (GIS), artificial intelligence/machine learning, and data analytics has considerably improved the accuracy and comprehensiveness of fire risk assessments.

More precisely, since the initiation of the first Earth Observation Satellite Mission, starting with NASA's Landsat-1 in 1972, satellite systems have continuously collected data, allowing for long-term observations and analysis of changes in vegetation, land use, fire events, and environmental conditions. These historical data enable researchers to study patterns, trends, and potential fire risks. Thenceforth, the combination of RS data with other spatial data in a GIS environment can be found in various studies [7–12] and has proved its value in fire risk assessment.

Fire behavior models are essential tools used in wildfire management and research to understand and predict the behavior of wildfires under various conditions. Instances of such models include FLAMMAP and FARSITE, which provide valuable insights into fire spread, flame length, and fireline intensity, considering factors like fuel type, topography, weather conditions, and wind patterns [13,14]. Additionally, BehavePlus is widely used for analyzing fire behavior in different fuel types and weather conditions [15], while WRF-SFIRE integrates weather and fire behavior modeling for high-resolution fire spread predictions [16]. Furthermore, Prometheus specializes in predicting wildfires in Mediterranean landscapes, and FireFamilyPlus offers a suite of models for simulating fire spread and behavior in different ecosystems [17]. FireLib, on the other hand, provides a library of fire behavior algorithms for various applications [18].

In parallel, AI/ML and data analytics are becoming increasingly valuable in assessing and managing fire risk, as they can analyze large volumes of data, extract patterns, and make predictions. These technologies have been widely adopted by the scientific community and have resulted in accurate and analysis-ready results [11,19,20].

Among the plethora of advances and developed methods, it is also worth mentioning the analytical network process (ANP) and the analytical hierarchical process (AHP). Both are under the umbrella of decision-making methods and are designed to handle complex and interconnected decision-making situations that involve interdependencies and feedback among criteria, alternatives, and stakeholders [8]. These methods are usually combined with GIS techniques and fuzzy logic and result in accurate field risk assessments [8,10,21]. Other studies investigated other types of more complex models, such as Kameleaon FireEx (KFX) Fire and Heat Transfer Simulations, which require special care due to assumptions and uncertainties, aimed to establish a methodology that could be applied to the global scale [22–24].

Apart from advancements and scientific trends in fire risk research, the literature also identifies limitations and unaddressed issues that need further investigation. More precisely, most studies do not reach a high-detail analysis of fire risk in terms of scale (vital for safeguarding human life) but mainly refer to areas of small extent and average

scale [25–27]. At the same time, several studies do not incorporate field knowledge [12], which, if coupled with scientific analysis, could offer very accurate and targeted results that could essentially support public actors when dealing with disasters. Last but not least, most of these research works, though very important, fail to produce hands-on results in terms of operational applicability, for example, able to be incorporated into a Decision Support System (DSS) that will directly be of help to civil protection services and first responders (e.g., fire department).

Through this applied research work, we attempt to meet these challenges, but most of all needs. Infusing the scientific state of the art, as identified in the literature, in a modular, scalable, and transferable methodology that focuses on identifying fire risk inside the neighborhoods of citizens, where it matters, and considering and recording the specific critical characteristics for each household, property, road, and part of the infrastructure leads to a complete, high-detail, contemporary and end-to-end fire risk management approach that starts from assessing the risk at a building-block level and achieves the design of evacuation plans and real-time management of extreme events.

This type of work has never been implemented in Greece, which is a country that faces extreme forest fire events almost every year, constantly rising in frequency and severity during the last two decades. In particular, Greece has witnessed numerous fire incidents with significant consequences [28]. Among recent incidents, the fire in Mati, Attica, commencing on 23 July 2018, resulted in the tragic loss of 102 human lives and substantial property damage over an area of 1300 hectares [29].

Considering the above, the Prefecture of Attica (Greece) answered the imminent need to counteract this constantly rising threat by financing the research project "Seismic, Fire Flood Risk Assessment in Attica Region, Greece". The presented methodology and outputs were developed and produced under this project. The Operational Unit "BEYOND Centre of Earth Observation Research and Satellite Remote Sensing", of the Institute for Astronomy, Astrophysics, Space Applications and Remote Sensing, of the National Observatory of Athens, is coordinating and leading the first part of this very significant project.

In detail, this study proposes an integrated approach that combines advanced techniques, tools, and datasets, such as artificial intelligence/machine learning (AI/ML), remote sensing (RS), geographic information systems (GIS), and field observations. The main focus of this work is to conduct a detailed fire risk assessment and propose efficient management plans (before and during the event) for a building-block-level analysis in selected peri-urban areas.

The methodology starts with the assessment of the vulnerability and the economic exposure (in terms of land value) of the studied area, followed by the prediction/identification of possible fire ignition points (through RS and AI/ML) and numerous simulations of the spatiotemporal spread of various fire events based on different scenarios of ignition points and wind characteristics (intensity, direction). The aforementioned combined analysis leads to high-resolution fire risk assessment maps that are also used to plan field campaigns, where critical areas and local high-risk characteristics are identified and recorded. The in situ assessments also work backward as a validation and update procedure of the risk maps. The final step is to design and propose localized and targeted management plans (e.g., evacuation routes, evacuation orders, refuge areas, fire protection zones, etc.), based on all the preceding work. It is worth noting that while this work incorporates several cutting-edge technologies and diverse datasets, it has also encountered certain limitations. These limitations encompass the use of outdated census data from 2011, data gaps, and constraints in creating the exposure layer. In response to these challenges, we leveraged a combination of field visits, remote sensing data, orthophoto maps, and land value information to address these impediments.

Finally, all the project's outputs will be integrated unto an interactive web platform that will act as an operational tool under a wider DSS that will assist regional and local authorities to efficiently fight forest fires threats to urban and peri-urban areas. All outputs

of the study area were presented to, discussed with, and evaluated by civil protection services, first responders, local authorities, and experts in the field.

## 2. Materials and Methods

### 2.1. Study Area

The Attica Region of Greece is a significant area that includes the capital city and plays a vital role in supporting a considerable portion (almost 50%) of the country's population and essential infrastructure. The region is known for its diverse and rich natural landscape, which includes numerous peri-urban settlements, as well as for its diverse topography, with hills and mountains.

The study area includes the settlements of Keratea and Kaki (or Kakia) Thalassa (Figure 1), which belong to the Municipality of Lavreotiki in the Attica Region. The average altitude of the area is approximately 187.13 m and the climate is warm Mediterranean with little rain, mild winters, and hot summers. The area experiences relatively little rainfall throughout the year, with the rainy season typically occurring between December and March. The overall annual rainfall does not exceed 409 mm, indicating a dry climate. The winters are generally mild, while the summers are hot and characterized by very high temperatures. During summer, temperatures can exceed 40 °C occasionally. The abundance of sunny days is a typical feature of this Mediterranean climate. Even in winter, the region experiences around 18 days per month that are sunny or partly cloudy. In the summer months, clear weather predominates, with approximately 24 days out of 30 being characterized by clear skies.

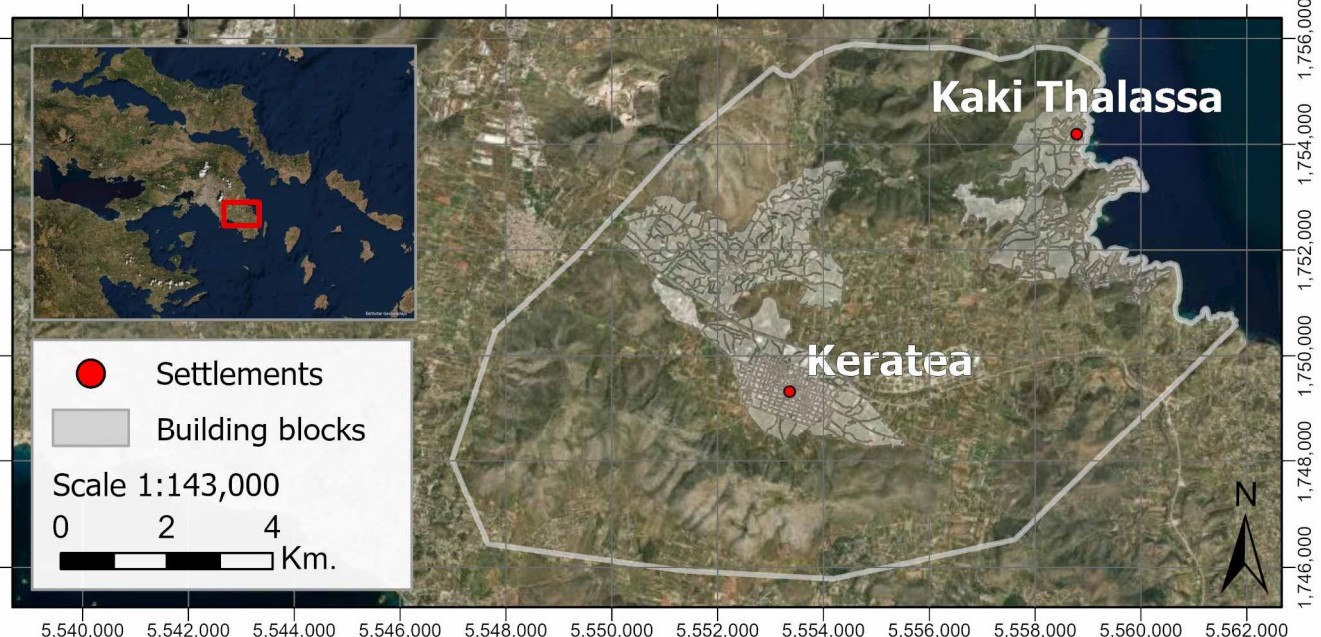

**Figure 1.** Study area.

### 2.2. Methodology

2.2.1. Overview

The methodology of this study involves the integration of two different models, namely BEYOND's Machine Learning model for daily fire risk forecasting [30] and the parameterized FlamMap fire spread model that is a combination of FlamMap [13] and FireHUB, which is an integrated early warning and near-real time monitoring system developed by the Operational Unit BEYOND of NOA [31]. Both of these models were adapted and calibrated to fit the local conditions and characteristics of the study area. Moreover, it offers the only high-detailed integrated approach in the Greek territory. In

particular, it stands out from other existing solutions by emphasizing the establishment of pre-event actions (e.g., road cleaning) to mitigate fire risk effectively.

The methodological framework was developed considering already established and applied methodologies EMSN041, EMSN059, and FireHub [30,32], and an extensive literature review [8,33,34]. Building upon this foundation, the methodology employed in this study utilizes a next-day fire risk model, which has been trained on historical data spanning from 2010 to 2018 and has been applied for daily predictions during the years 2019 to 2021, enabling precise assessment of ignition hazards within 100 × 100 m subareas of the area of interest (AOI). Furthermore, an integrated custom Flamap model is employed to simulate potential wildfire behavior originating from locations identified by the fire risk model. To ensure the methodology's reliability, real-world validation is conducted through on-site visits, during which specific risks such as homes near forests with wooden structures, overgrown yards, power line/tree branch intersections, and poorly maintained roads with steep inclines and dead-end sections are systematically identified. This holistic approach not only enhances the robustness of the methodology but also significantly contributes to more effective wildfire risk assessment and mitigation strategies. The flowchart of the methodology is presented in Figure 2.

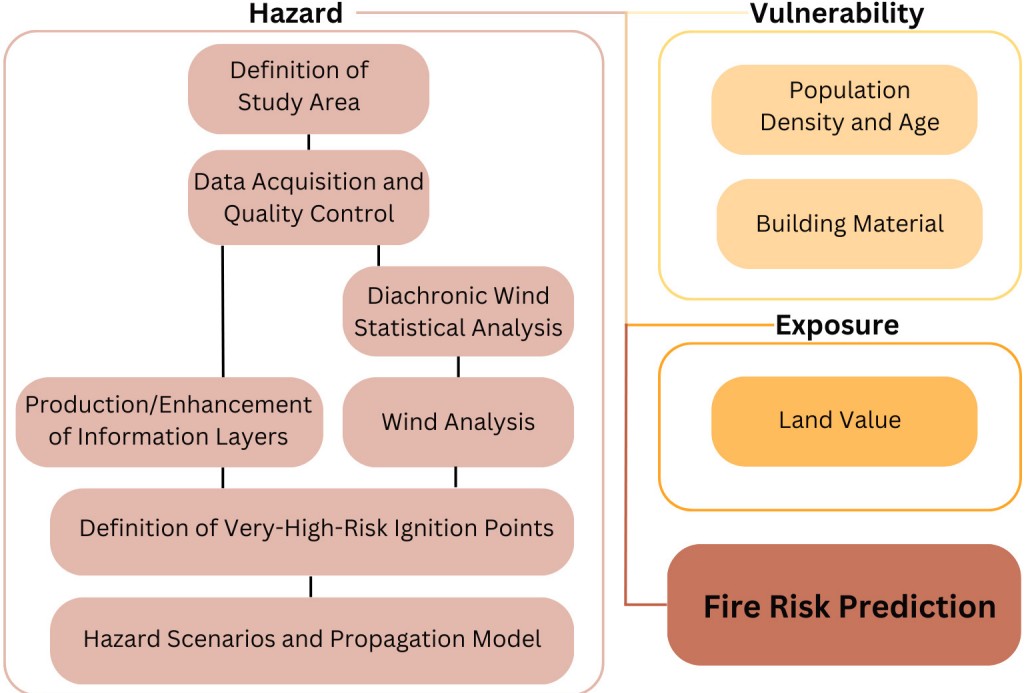

**Figure 2.** Flowchart of the methodology.

In a nutshell, various simulations of spatio-temporal fire spread were performed for the area of interest. The worst-case fire scenarios, in terms of time and extent propagation, were selected and combined to render the overall hazard for the area. In parallel, the region's vulnerability was composed and classified (into 5 classes from very low to very high according to the Natural Breaks classifier) by co-assessing the population (density and age) and building characteristics (construction material) of the area of interest. In addition, the area's land value zones (also categorized into 5 classes) were used as a qualitative exposure proxy of the possible economic impact on the region in case of fire. The coupling of the above produced a high-detail (building-block-level) fire-risk map of the area, based on which the high-risk areas were identified and visited through exhaustive field campaigns. The knowledge acquired from the field visits helped the iterative process of validating and updating the risk maps, the identification of critical points and important

sites, and finally resulted in the development of operational management plans and fire prevention suggestions.

### 2.2.2. Analysis of Wind Characteristics

For the analysis of wind characteristics in the study area, wind velocity and direction data were collected from two sources: the ERA-5 mission [35] covering the period from 1979 to 2021 and the METEO platform [36] for the ground meteorological station of Lavrio for the period 2009 to 2021. The ERA-5 Land reanalysis data provided the $U$ and $V$ wind components on an hourly basis, offering a spatial resolution of 9 km. Wind speed was calculated using the Pythagoras theorem [37]:

$$WS = \sqrt{U^2 + V^2} \tag{1}$$

where $WS$ is the wind speed, $U$ is the horizontal speed of air moving towards the east, at a height of ten meters above the surface of the Earth, in meters per second and $V$ is the horizontal speed of air moving towards the north, at a height of ten meters above the surface of the Earth, in meters per second.

Wind direction ($\theta$) was determined by applying the following:

$$\theta = \text{atan2}(V/U) \tag{2}$$

All these calculations and subsequent analyses were performed utilizing the Python 3.10.6 programming environment.

The METEO ground station time series data can effectively capture actual weather conditions within the study area. In this dataset, daily averages and maximum wind intensities are recorded, along with prevailing wind directions.

Due to the limited operational time of the meteorological station in the area, a historical statistical wind analysis for the whole region was performed by incorporating the large timeseries of the gridded ERA-5 data and the observed wind conditions on a local-scale. The data from ERA-5 and METEO were integrated using a quantile mapping technique [38]. This merging process ensured that the gridded data were aligned with the distribution of point observations from the meteorological station at Lavrio, thus enhancing the overall reliability and accuracy of the wind data on a finer spatial and temporal scale.

### 2.2.3. Determination of Fire Ignition Points

Various datasets were employed to evaluate and analyze potential fire ignition points. These datasets include the Digital Elevation Model (DEM) and its related derivatives such as aspect and slope, ERA-5, and METEO data capturing temperature, wind characteristics (velocity and direction), and rainfall, as well as the adjusted Corine Land Use/Land Cover product from the Copernicus Initiative. Additionally, satellite-based spectral indices like NDVI and EVI were incorporated. Selecting appropriate ignition points is a critical aspect of simulating fire scenarios, as it allows for the assessment of potential fire spread patterns and their potential impact on residential areas.

For this purpose, BEYOND's Machine Learning model of daily fire risk forecasting was utilized [30]. In order to study the worst-case wind scenarios in relation to potential wildfires in the area, classes were created based on wind intensity, and extreme wind intensities were examined in conjunction with prevailing directions. As evident from the wind rose diagram (Figure 3), in this region, north, northwest, and northeast winds dominate, while wind intensities greater than 7 Beaufort often include south and southeast winds. An important observation is the absence of east winds with intensities exceeding 6 Beaufort. For various wildfire spread scenarios and for each ignition point, only those directions that could potentially threaten settlements and points of high interest were considered.

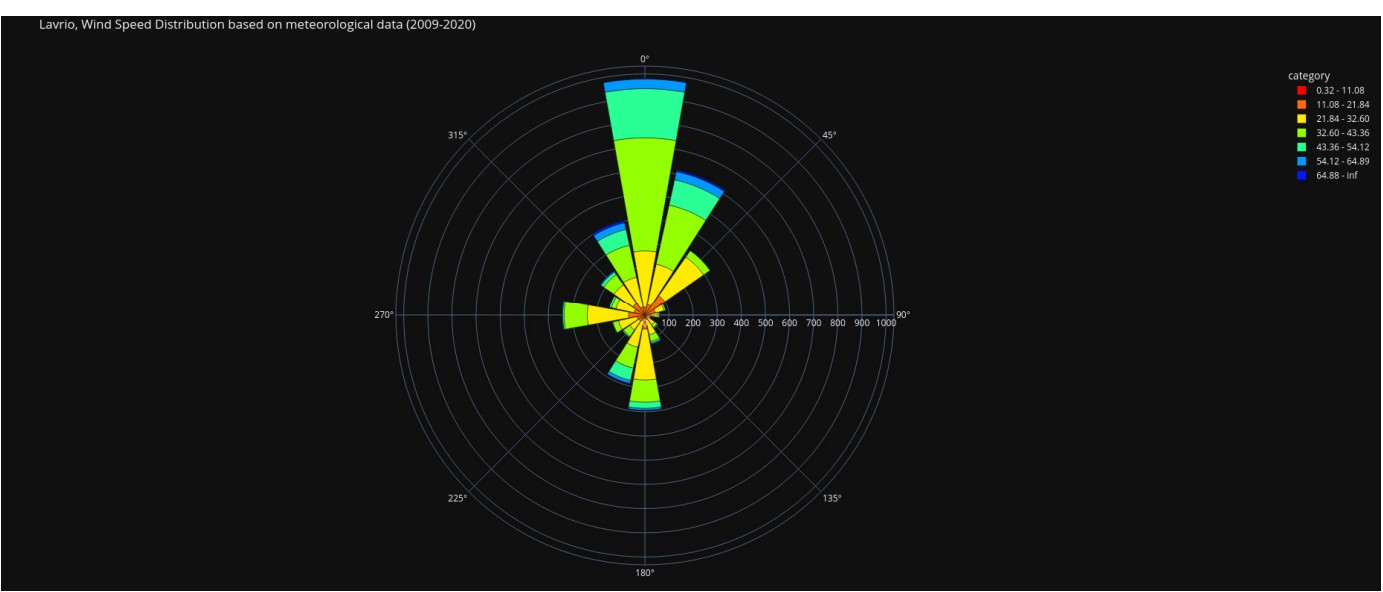

**Figure 3.** Wind rose of the dominant wind directions and intensities.

### 2.2.4. Hazard Scenarios

FlamMap 6.2 (United States Forest Service) is a fire behavior analysis software that computes the potential fire behavior characteristics such as spread rate, flame length, and fireline intensity under constant environmental conditions at a specific time over the entire landscape (LCP) using local weather and fuel moisture conditions and the local topographic layers. The creation of the LCP file requires topographic data, fire behavior fuel models, and forest canopy characteristics such as cover, height, base height, and bulk density to simulate the surface fire spread, while also including the possibility of a crown fire. In addition to these layers, information on dead and live fuel moisture, weather conditions, and wind speed and direction are necessary inputs for accurate fire behavior simulations [39].

However, at least five geospatial layers, those of elevation, slope, aspect, fuel model, and canopy cover, could be inserted as inputs in order to simulate the surface fire spread, along with the location of ignition points, weather data, and wind direction and intensity. In this study, FlamMap was modified to function in a Linux environment and was translated into the Python programming language. The necessary input layers were used, meaning the Digital Elevation Model (DEM), aspect, slope, fuel map, and canopy bulk density (as depicted in Figure 4). This streamlined configuration improves the time efficiency of fire behavior simulations while retaining the crucial data needed for precise and reliable results.

With regard to the data sources, the DEM was derived from orthophoto maps from the Hellenic Cadastre with a spatial resolution of 5 m and was utilized to generate the aspect and slope (in degrees) layers through GIS. In this study, the surface fuel, the fuel's moisture content, and the vegetation characteristics of the area were determined using an adjusted land use/cover map via GIS, high-resolution satellite imagery, and photo interpretation. Each fuel model was assigned to a vegetation type as a combination of custom and standard fuel models based on the research work of [40,41].

More specifically, code 18 refers to forests and shrublands, code 22 represents permanent crops, 24 is the non-fuels and 25 corresponds to the grasslands and annual crops (Figure 3). Considering the density layer, it represents the amount of available fuel per unit volume of the surface covered by the canopy space. It is a crucial factor that affects the probability of a fire outbreak and the rate of crown fire spread. This layer was categorized into 4 classes (0 $m^3$, up to 100 $m^3$, 100–300 $m^3$, and above 300 $m^3$) for the area of interest, referring to the estimated timber volume of the available fuel per $m^3$.

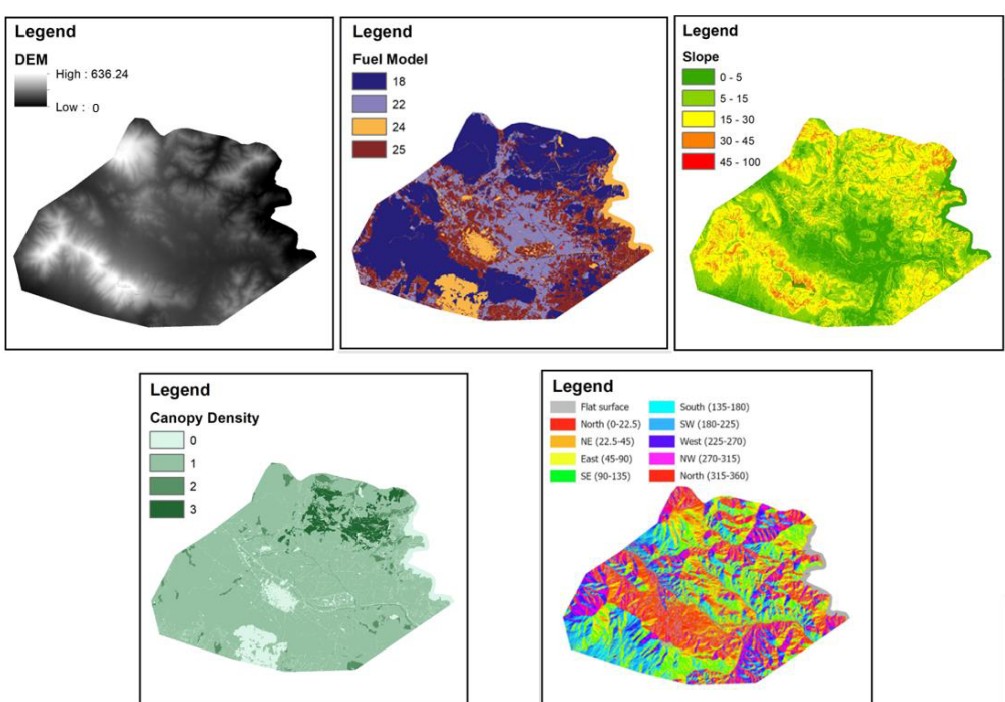

**Figure 4.** Input layers for parameterized FlamMap.

### 2.2.5. Vulnerability

Assessing the vulnerability of a study area is a crucial step in understanding its resilience to potential hazards, such as fire incidents. In this study, focus was given to three critical layers to assess fire vulnerability: population density, population age, and building materials [27]. Population density and age are essential factors in fire risk assessment because they influence the number of people potentially affected by a fire event.

Higher population density may lead to more significant impacts in terms of evacuation, emergency response, and the overall potential for casualties [42,43]. Additionally, considering population age helps identify vulnerable groups, such as the elderly or children, who may require special attention and assistance during fire emergencies [44]. The choice of building materials as a vulnerability factor is also important as it directly impacts the flammability and combustibility of structures [45]. Buildings constructed with certain materials may be more susceptible to fire hazards, and this information is crucial for understanding the potential spread and intensity of fires within the study area. These layers were created using data from the 2011 census obtained from the Hellenic Statistical Authority (HSA). The census data were processed and analyzed (quality check and improvement) with the support of high-resolution satellite images and orthophoto maps to ensure accurate and reliable results.

#### Population Density

To evaluate the vulnerability related to population density, this study utilized detailed census data at the building-block level to determine the total number of people residing permanently within the study area's settlements. By dividing this population count by the total area of each building block, the population density was calculated in units of persons per square meter using Equation (3).

$$Population\ density = \frac{Total\ population\ (per\ building\ block)}{Total\ area\ (per\ building\ block)} \tag{3}$$

Once the population density layer was generated, a classification process was employed to divide the population density layer into five vulnerability classes. For this

purpose, the natural breaks classifier was applied. Specifically, the natural breaks classifier is a statistical method that seeks to identify natural groupings in data. This classifier helps determine appropriate class intervals based on the distribution of population density values [46]. The output of this process was further enhanced and updated (where needed) by utilizing the notes and data derived during the field visits.

By enhancing population density estimation with field observations and recordings at a building-block level, the vulnerability assessment becomes more refined and representative of the specific characteristics of the study area. The approach accounts for variations in population distribution and the concentration of individuals in different parts of the study site, providing a more accurate and custom-made depiction of vulnerability related to population density.

Population Age

The study's analysis of census data to understand the vulnerability of different age groups to fire incidents is a crucial aspect of the fire risk assessment [44]. The census data provide age information for the following age groups: 0–9, 10–19, . . ., 60–69, 70–79, 80 plus. By grouping age data into five categories [Table 1] based on similarities in evacuation capacity during fire events, the analysis was simplified and supported the identification of the age groups that might have greater difficulty coping with fire incidents.

**Table 1.** Age groups and assigned weights.

| Ages | Age Group | Weights |
| --- | --- | --- |
| 20 to 39 | 1 | 0.05 |
| 40 to 49 | 2 | 0.1 |
| 50 to 59 | 3 | 0.15 |
| 10 to 19 and 60 to 69 | 4 | 0.25 |
| 0 to 9 and >70 | 5 | 0.45 |

(Age group vulnerability, 1: very low, 2: low, 3: medium 4: high and 5: very high).

By assigning weights to each age group based on their vulnerability and coping capacity in fire events, it is easier to quantify the relative importance of different age groups in the vulnerability assessment. To calculate the weighted population value for each age group in a building block, the total population of that age group was multiplied by its corresponding weight.

To ensure accurate comparisons across different areas, the results were standardized by dividing the weighted population of each age group by the total population in the specific building block, as presented in Equation (4).

$$Age\ Vulnerability = \frac{\sum_{k=0}^{k=5}(Age\ group(k) * Weighting\ factor(k))}{Total\ population\ per\ building\ block} \qquad (4)$$

Building Characteristics

The consideration of building characteristics is crucial in assessing vulnerability and fire resistance. Building materials strongly influence a building's susceptibility to fire, how quickly it can ignite, and how fire might spread within the structure [45]. In this study, the census data provided valuable information about the materials that were used in the buildings within the study area. In particular, the materials that can be found in the study site are wood, metal, other materials, bricks/concrete, and stones. It is noteworthy that the 'other materials' of the census data were undefined.

By categorizing the building blocks based on the materials of the buildings situated in each one of them, it was feasible to identify different levels of vulnerability. Building blocks that had at least one building made of wood were classified as very high in terms of vulnerability (class 5). This classification is appropriate because wood is highly flammable

and can contribute to rapid fire spread, making these building blocks more susceptible to fire events.

On the other hand, building blocks consisting solely of buildings made of stone were assigned to the very low vulnerability class (class 1). Stone buildings are less likely to ignite and have better fire resistance properties compared to wood. As a result, these building blocks are less vulnerable to fire incidents, providing a higher level of resilience in case of fire events.

In addition, building blocks that had at least one building made of metal or other materials were classified as high (class 4) and medium (class 3), respectively. This stems from the fact that metal and other materials (as identified and recorded in the field visits) are quite flammable.

The building blocks that had at least one building made of brick and/or concrete were classified as low (class 2) as they are characterized by fire resistance [45]. The criteria of the classification of the building materials were based on literature research [45] and on expert (e.g., civil engineers) knowledge. Thence, the final building material vulnerability map was enhanced and updated with the field notes and gathered data.

This approach allows for a simplified but effective assessment of building vulnerability based on construction materials [47]. It helps in identifying areas with higher fire risk due to the presence of flammable materials like wood and areas with better fire resistance properties due to the prevalence of resilient constructions. Building characteristics were updated by field observations.

Total Vulnerability

The total vulnerability was estimated through the spatial integration (unweighted cartographic overlay) of the aforementioned classified vulnerability layers. Firstly, the population density was combined with the population age classified layer, and thence, the resulting layer was combined with the building material vulnerability. Specifically:

1.  This study combined population density and population age vulnerability layers to assess human vulnerability to fire incidents in the area of interest. This accounted for different demographic characteristics affecting vulnerability in the study area.
2.  The resulting layer was then combined with the building material vulnerability to estimate the total vulnerability of each building block by applying equal weights to population density, population age, and building material for assessing the vulnerability to fire risks.

2.2.6. Land Value Exposure

The creation of the exposure layer in terms of land value is a vital facet of the fire risk assessment, as it helps understand the potential economic impact in the event of a fire. The exposure layer was developed by combining data on land use, land values in euros per square meter (EUR/m$^2$), very-high-resolution satellite images, and orthophoto maps. The land value dataset, obtained from the Greek Ministry of Digital Government, provides valuable information on the spatial distribution of land values within the study area.

To make a qualitative estimation of the potential financial cost associated with fire incidents, the land value zones were classified into five classes. The classification involved grouping the zones based on their respective land values. The lower classes represent areas that are likely to experience lower economic damages (less total housing price) in case of a fire incident, while the higher classes represent zones where financial damages would be more significant if a fire were to occur.

By categorizing the land value zones into classes (<1000 EUR/m$^2$, 1000–1300, 1300–1600, 1600–2000, and >2000 EUR/m$^2$), the exposure layer provides an educated understanding of the potential economic risks and consequences associated with fire incidents across different areas within the study site. The final exposure layer was enhanced and updated by the field observations. Field recordings provide valuable ground-truth information about high-risk buildings, in terms of land value, critical infrastructure, hotels, etc., which

can significantly impact the potential economic risks and consequences associated with fire incidents.

2.2.7. Fire Risk

For the estimation of fire risk, the total hazard, total vulnerability, and exposure layers were combined with equal importance via weighted cartographic overlay. Each of these layers represents different critical aspects of fire risk, and their integration allows for a comprehensive and holistic understanding of the overall fire risk landscape in the study area.

The hazard layer provides information about the potential fire intensity, spread, and likelihood in different regions. It takes into account factors such as vegetation, topography, and climate conditions that contribute to fire ignition and propagation. Understanding the hazard layer is crucial for identifying areas at higher risk of fire occurrence.

The total vulnerability layer incorporates multiple factors, such as population density, age distribution, and building materials, to assess the vulnerability of the study area to fire incidents. By considering both human and structural vulnerabilities, this layer highlights areas where the impact of fire events could be more severe due to the characteristics of the population and buildings.

The exposure layer provides insights into the potential financial risk associated with fire incidents by considering land values and the distribution of valuable assets. This layer helps identify areas with higher economic value and potential losses in case of fire events.

## 3. Results

### 3.1. Hazard Scenarios

Each worst-case fire spread simulation represents a specific scenario with unique characteristics, such as the ignition point, the fire spread, and wind conditions. Spatially combining these individual worst-case fire spread simulations allows the creation of a unified comprehensive representation of the overall worst-case fire hazard across the entire area of interest.

The left side of Figure 5 provides a clear visual representation of the fire simulation conducted under the following specific conditions: a west wind with an intensity of 9 on the Beaufort scale (Bf) and an ignition point located west of the settlement of Kaki Thalassa. The classes labeled "Very High" to "Low" correspond to different time intervals depicting the progression of the fire spread. The "Very High" class indicates the initial hour following the ignition of the fire event, and within this hour, the fire has already penetrated the settlement. The subsequent classes, "High", "Medium", and "Low", represent the fire's spread at up to 3 h, up to 6 h, and up to 10 h from the ignition of the fire event, respectively.

The right side of Figure 5 displays the visual representation of the fire simulation conducted under the following specific conditions: a north wind with an intensity of 9 on the Beaufort scale (Bf) and an ignition point situated on the northern side of the settlement of Kaki Thalassa. Similar to the simulation with the west wind, the fire once again penetrated the settlement within the first hour following the fire ignition event. These results underscore the swift and significant impact of the fire under the specified wind and ignition conditions, emphasizing the importance of timely intervention and preparedness measures to mitigate potential risks.

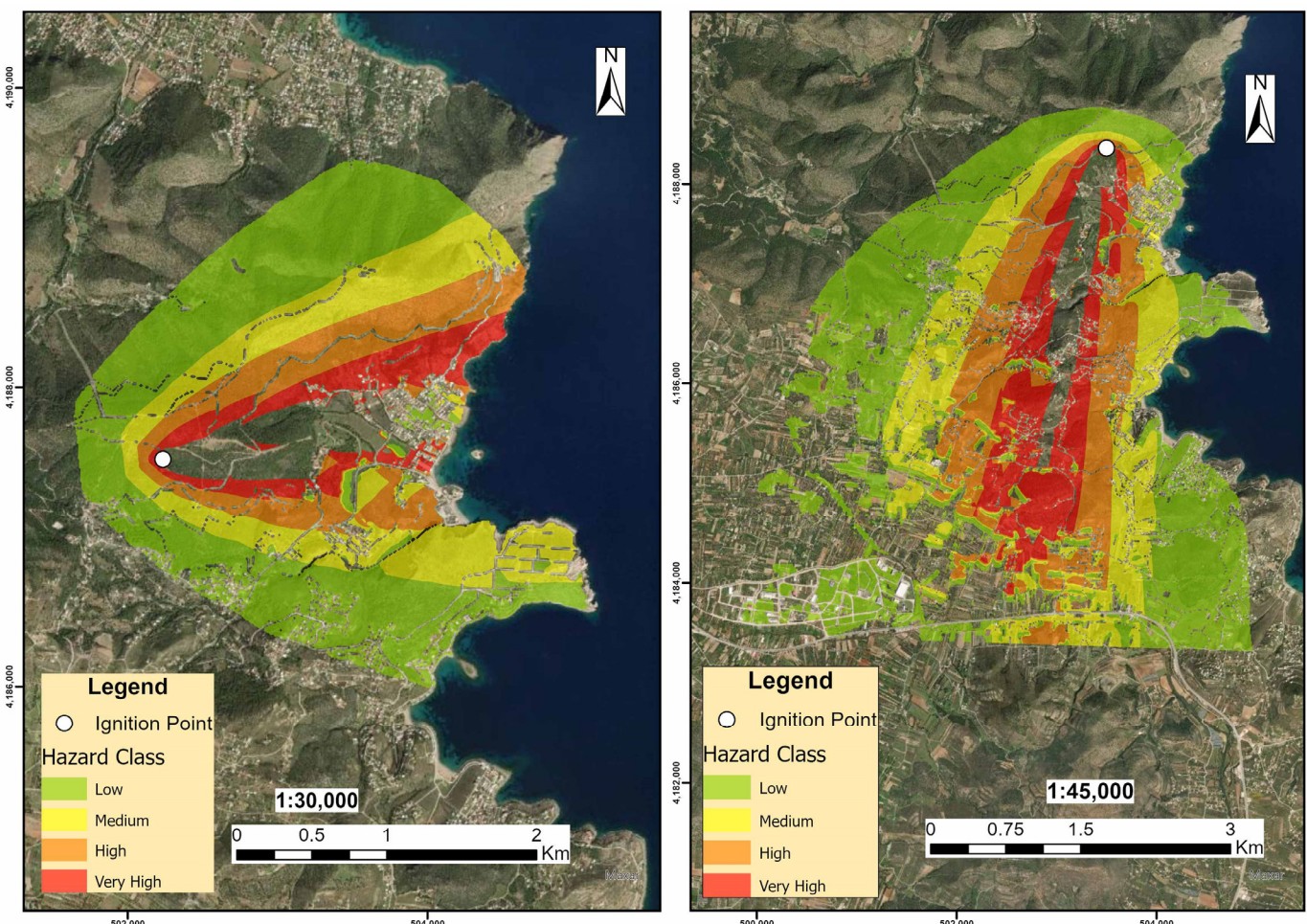

**Figure 5.** Examples of parameterized FlamMap.

### 3.2. Total Vulnerability

By integrating these vulnerability layers, decision-makers and emergency responders gain valuable insights into areas with higher overall vulnerability, which can aid in prioritizing fire risk mitigation strategies and emergency response planning.

The total vulnerability (Figure 6) in the northern, western, and southern parts of the settlement has been identified as high, primarily because of the presence of individuals with age-related limitations in case of evacuation. In light of this, providing assistance to these areas should be considered a high priority.

Building blocks near the coastal zone are identified as vulnerable due to the substantial influx of people during the summer months, leading to increased population density and potential risks during fire events. The reason of the increment in the population during the summer months is the existence of vacation properties in the broader area.

In the northern part of the settlement, certain building blocks are also identified as vulnerable due to the presence of flammable building materials, which, when combined with the presence of age groups that require assistance, result in their classification as very vulnerable areas.

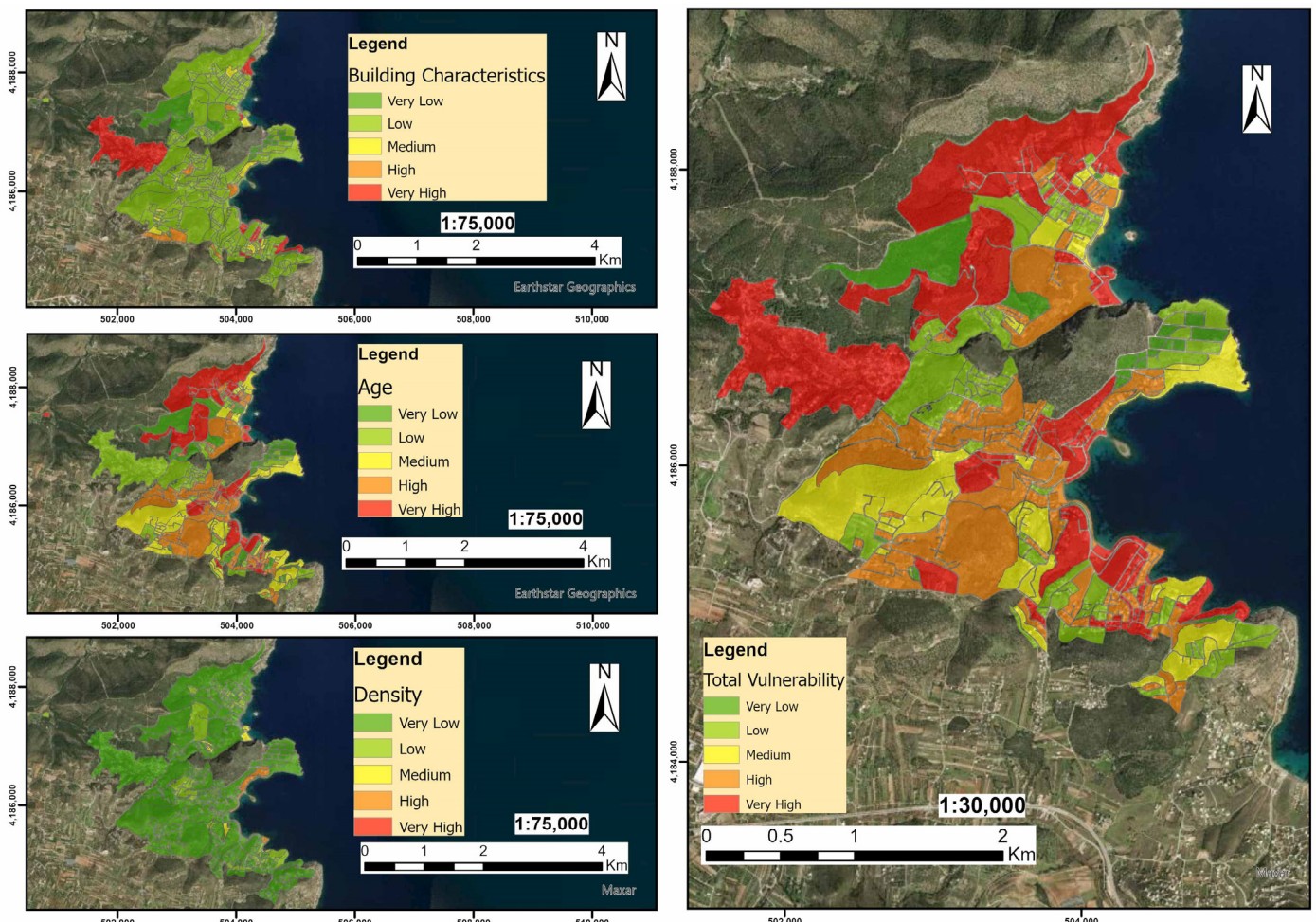

**Figure 6.** Total vulnerability estimation based on census data.

Finally, the total vulnerability values of some building blocks were modified based on the field data. As such, building blocks that include hospitals, churches, primary schools, and/or other critical infrastructure with a highly vulnerable population (e.g., elderly people) were characterized as very vulnerable (class 5).

### 3.3. Exposure

The exposure of land values in the Attica region were categorized into five classes (Figure 7) as follows: very low exposure (up to 1000 EUR/m$^2$), low exposure (1000–1300 EUR/m$^2$), medium exposure (1300–1600 EUR/m$^2$), high exposure (1600–2000 EUR/m$^2$), and very high exposure (above 2000 EUR/m$^2$). The studied area is primarily characterized as having a medium exposure, with land values not exceeding 1600 EUR/m$^2$.

However, it is noteworthy that Kaki Thalassa has building blocks with higher land values than the majority of the building blocks of the broader area. As such, it deserves higher priority in terms of financial aspects.

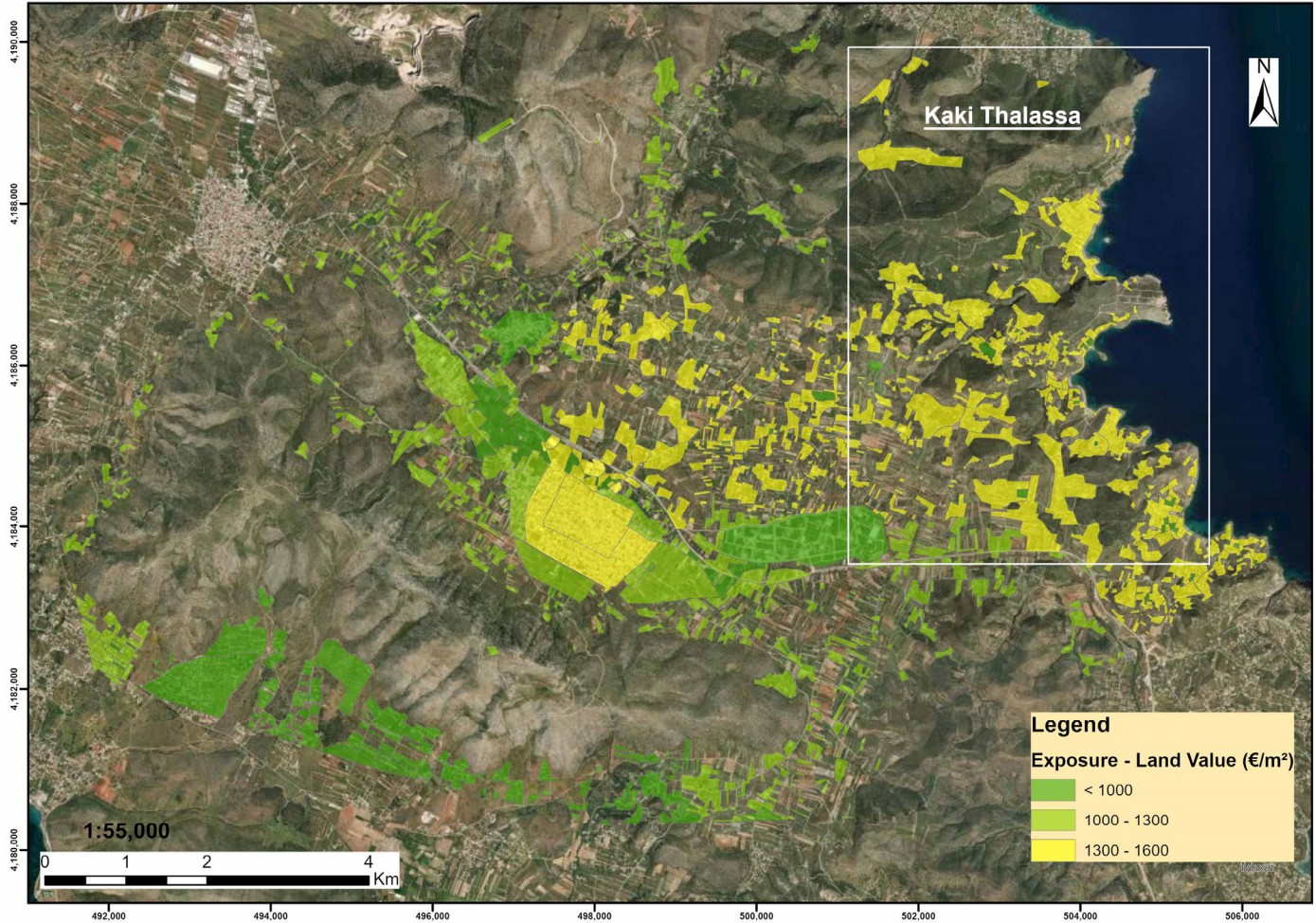

**Figure 7.** Exposure layer.

*3.4. Fire Risk Maps*

In Figure 8, a substantial portion of the area has been identified as being at high risk due to its close proximity to forested regions, particularly in the northern part. This geographical arrangement creates a direct pathway for the fire to penetrate the settlement, posing a significant threat to buildings and infrastructure. In addition, the road network in the area consists of low-quality roads with steep slopes and dead ends all over the area, hindering quick evacuation during a fire event as there is only one main road to exit the settlement, which is expected to have congestion in case of evacuation.

Moreover, the coastal area, known for its high population gathering due to the presence of a church, restaurants, and beach bars, also serves as a potential high-risk hotspot in case of fire incidents. The assessment has identified two open spaces (a parking area and a coastal zone) situated on the northwest coast, which have been identified as assembly points in case of an emergency.

Finally, the presence of water tanks in the area of Kaki Thalassa provides a valuable resource for firefighting efforts during a fire event. However, the absence of fire outposts, police stations, and other critical infrastructure does highlight some potential challenges that should be addressed in future fire risk mitigation measures.

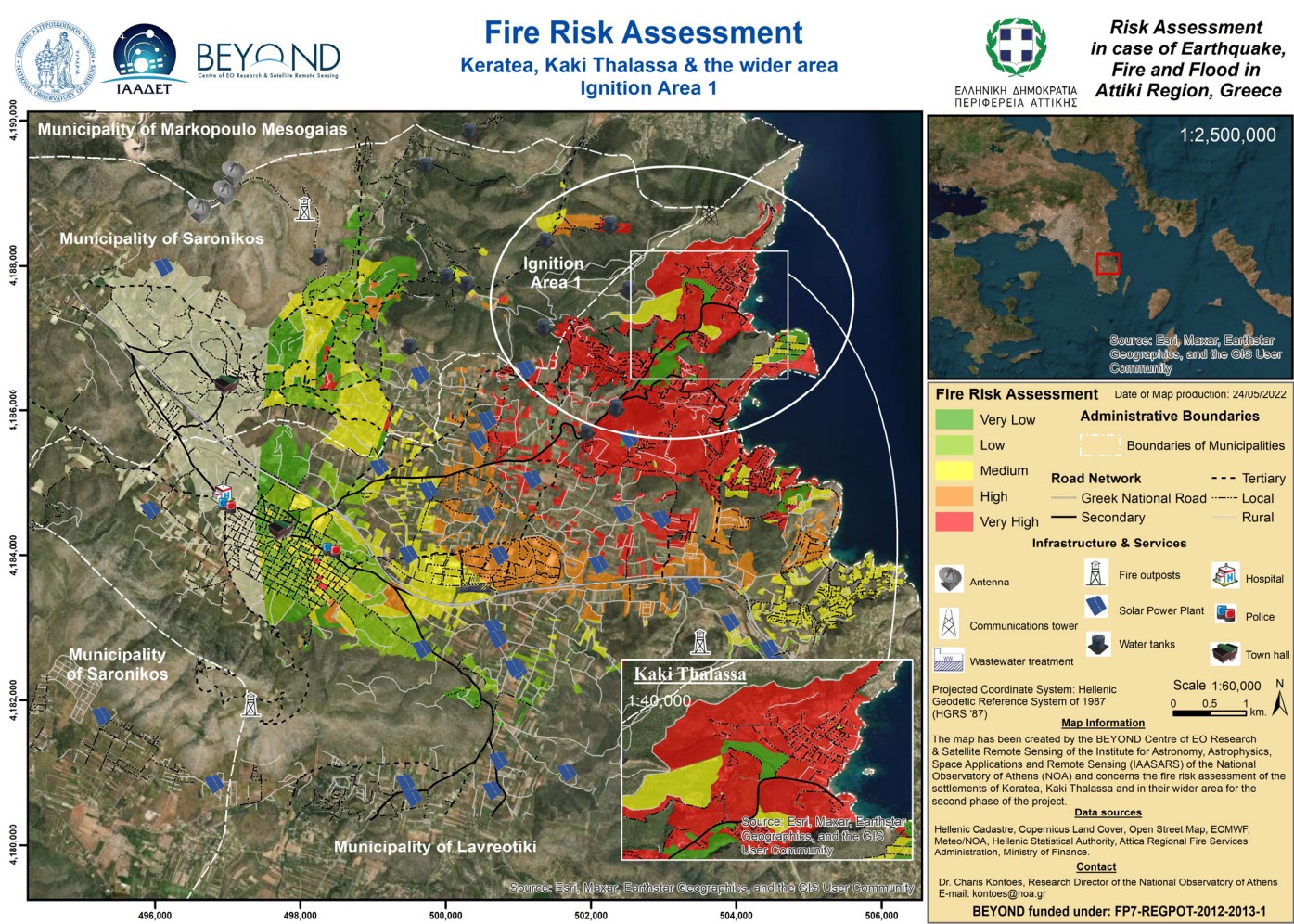

**Figure 8.** Fire risk assessment for Kaki Thalassa.

### 3.5. Field Campaigns and Management Planning

The settlement of Kaki Thalassa presents unique challenges for fire risk management due to its radial arrangement along the Keratea–Kaki Thalassa Provincial Road and its positioning on the slope of the Mavrinora hill, which is located on the west side of the settlement. During the fieldwork, critical characteristics were recorded, such as dead ends all over the settlement as it is radially arranged, steep roads mainly in the northern part, and the presence of fire protection infrastructure at the end of the main road Areos (northern part) and in the south, close to the sea, near the parking area.

Based on the fire risk maps and field observations, evacuation maps were created for the defined ignition areas. These maps identify the assembly points in case of emergencies and the congestion points that are expected on the two parallel roads (Areos and Poseidonos) of the settlement that lead to the main road Keratea Provincial Road——Kaki Thalassa. Further enhancement of fire prevention and safety can be achieved through the installation of fire alarms and protection systems in buildings and properties, ensuring early detection of fire incidents. Additionally, cleaning roads and properties from debris, pine needles, and waste can reduce fire fuel and create fire breaks, limiting the potential spread of fires.

By incorporating these measures, the settlement of Kaki Thalassa can improve its fire resilience and preparedness.

Proposed Evacuation Plan

The evacuation plan for the settlement of Kaki Thalassa is a critical component of fire risk management, ensuring the safety of residents during potential fire incidents. The plan considers both pedestrian and vehicular evacuation routes to facilitate the swift and efficient movement of people to safe areas.

More precisely, the main escape routes for pedestrians towards the sea (Posidonos Street) are Apollonos Street, Delphi Street, and Kaki Thalassas Avenue–Kerateas. These routes are strategically identified to allow residents to move away from the fire-affected areas and reach the shoreline, where they can potentially evacuate by watercraft if necessary.

Additionally, Figure 9a illustrates the suggested gathering areas, which include an open parking area and an adjacent open space near the shore. These designated gathering points provide safe assembly areas for the pedestrian population, enabling an organized evacuation process and efficient coordination during emergencies. The proximity to the shoreline allows for potential evacuation by watercraft, providing an additional means of escape if the need arises.

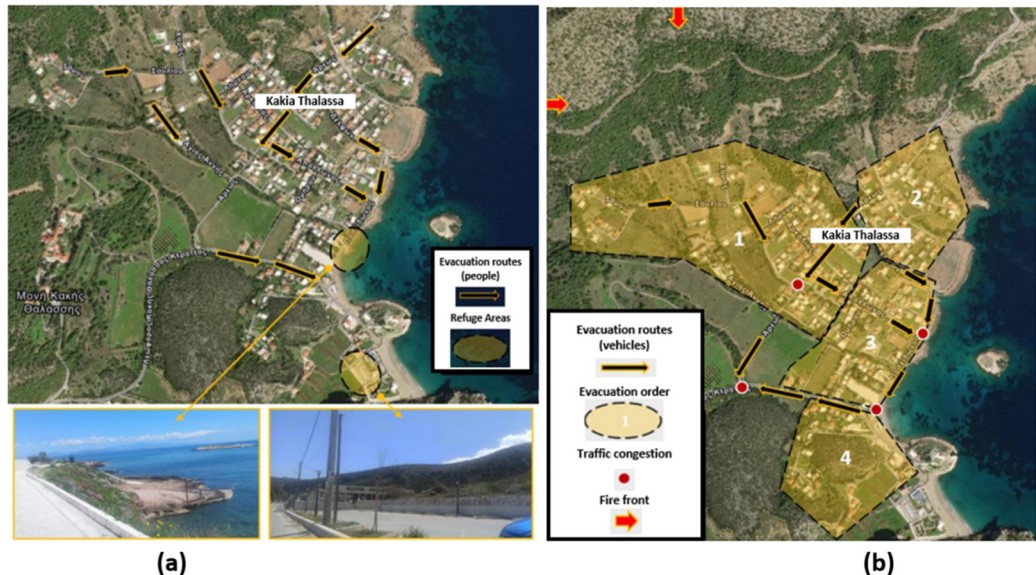

**Figure 9.** Evacuation map for Kaki Thalassa. (**a**) Evacuation route for pedestrians and refuge areas, (**b**) evacuation routes for vehicles along with evacuation order, traffic congestion points, and fire front.

Moreover, Figure 9b depicts the evacuation of vehicles that will occur via Kaki Thalassa–Keratea Avenue, heading towards Keratea. Priority will be given to the houses located northwest of the settlement, as they are closer to potential fire ignition areas and may face immediate threat. Subsequently, the rest of the area will follow the evacuation plan, ensuring a systematic and orderly departure of vehicles from the settlement.

Furthermore, it is anticipated that congestion may occur at the intersection of Kaki Thalassa–Keratea Avenue with Areos and Poseidonos Beach Street towards the Avenue. To prevent congestion and ensure a timely and effective evacuation process, the immediate evacuation of the settlement is prioritized, beginning simultaneously with the start of the fire. This early evacuation strategy aims to disperse traffic flow and avoid bottlenecks, allowing for smoother movement of vehicles and ensuring that residents can evacuate safely and quickly.

Overall, the evacuation plan takes into account both pedestrian and vehicular routes, as well as gathering areas and potential challenges in traffic management. By implementing this plan, the settlement of Kaki Thalassa can enhance its preparedness and responsiveness to fire incidents, safeguarding the lives and well-being of its residents during emergencies. The coordination of residents, local authorities, and emergency responders is crucial to

ensure the successful execution of the evacuation plan and mitigate the potential impact of fire hazards on the community.

## 4. Discussion

The proposed methodological framework presented in this study has demonstrated significant results, particularly when analyzing at the building-block level. However, like any comprehensive study, especially in complex fields like fire risk assessment, various challenges, limitations, and assumptions were encountered during its implementation. These key factors include data availability and quality, model limitations, the spatial and temporal resolution of datasets, accuracy assessment and validation, assumptions and simplifications made during the analysis, accessibility of digital information and field data, reliance on expert judgments, and the generalizability of the findings. Addressing these issues is crucial to enhance the robustness and applicability of the fire risk assessment methodology.

The assessment of fire behavior plays a pivotal role in the strategic planning and effective management of wildfires. It provides critical insights into the potential spread and intensity of fires, enabling decision-makers to devise appropriate strategies to mitigate risks and protect communities and resources. In this context, two widely employed wildfire simulation models, FARSITE and FlamMap, stand out as valuable tools for comprehending and predicting fire behavior under diverse conditions [14,48,49]. Jahdi et al. [50] calibrated the FARSITE fire spread model for wildfire simulation in northern Iranian forests using site-specific fuel models. The simulation results accurately replicated observed fire perimeters and behavior, particularly with the Scott and Burgan (2005) [40] standard fuel model.

Another study [50] demonstrated the model's high potential for estimating spatial variability in fire spread and behavior, providing valuable insights for wildfire risk monitoring and management in the region. FARSITE offers a comprehensive approach by simulating both surface and crown fire behavior, allowing for a more holistic understanding of fire dynamics [50]. On the other hand, FlamMap primarily focuses on surface fire behavior, making it more suitable for smaller-scale assessments and tactical fire management. FAR-SITE, with its capacity for large-scale fire spread simulations over complex terrain and weather conditions, excels in addressing the challenges of wildfires at broader spatial scales. Both models complement each other, and their combined use provides a comprehensive toolkit for fire managers and planners to effectively address the complex challenges posed by wildfires.

Although the FlamMap model is well accepted in the literature [51–53], it is unable to simulate fire scenarios where changes in wind direction and intensity occur. To address this limitation, multiple simulations based on various combinations of ignition points and wind conditions (direction and speed), were performed using the customized FlamMap. This approach ensured a more realistic understanding of fire behavior and could possibly enable decision-makers to implement targeted and effective wildfire management strategies to protect communities and resources.

With regard to the facet of data availability and quality, the literature review provides valuable insights into the types of data that can enhance the accuracy and comprehensiveness of fire risk assessments. Some of these data, as mentioned, include more detailed economic information [54], distance and visibility from fire watch towers [54], proximity to water and roads [54], road networks [55], unoccupied housing units [56], military ground information [56], and the existence of fire hydrants [57]. We plan to incorporate most of the aforementioned parameters (depending on data availability) into future advancements of the methodology.

The incorporation of data on population and building characteristics plays a crucial role in comprehending the potential impact of fire incidents and guiding targeted fire prevention strategies [54,56]. By analyzing demographic and socioeconomic information, researchers can gain insights into the vulnerability of different population segments to fire hazards [54–56]. For instance, certain age groups or socioeconomic categories may face

higher risks during fire events, necessitating the identification of vulnerable groups for targeted intervention strategies and emergency response plans [44,58]. However, in this study, the use of outdated census data from 2011 poses a limitation, as significant changes in population distribution and building characteristics might have occurred since then. To address this limitation and ensure more accurate assessments in the future, the plan involves updating the data with more recent census information.

Including data from other studies, such as population demographics and socioeconomic information, can bring valuable insights to fire risk assessments. Unfortunately, data availability for the Greek territory posed a challenge, and as a result, this research did not integrate this specific type of data into the fire risk assessment.

Furthermore, handling missing data is a common challenge in data-driven research, and it requires innovative techniques and alternative data sources to improve the completeness and accuracy of the dataset, particularly in areas with limited accessibility or missing data from the sources used in the study. In this study, the use of satellite imagery and remote sensing data proved valuable in filling the gaps and enhancing the spatial information of the study area. By employing these techniques, significant insights into land use, building characteristics, and population density were gained, particularly in areas with limited accessibility or missing data from the used sources.

Moreover, the fire exposure layer is a crucial component of fire risk assessment that combines land value and land use data with remote sensing information. This layer provides valuable insights into the potential impact of fire incidents on different areas based on their economic and land use characteristics. By integrating land value data (EUR/m$^2$) with land use information, decision-makers can better understand the potential financial losses and resource allocation required in the event of a fire. In cases where there is missing land value information for certain building blocks, neighboring data were leveraged to estimate the missing values [59]. This study employed an approach that involved using landscape characteristics derived from remote sensing data and the land value information of the closest neighboring building blocks with available data.

Nevertheless, a comprehensive and accurate assessment of the financial aspect of fire risk necessitates the integration of multiple datasets encompassing various economic factors. These datasets typically include information such as gross domestic product (GDP), reconstruction/repair costs, housing insurance data, financial expenses during the reconstruction period (including indirect expenses), tourism statistics, and business and industry data, among others.

Unfortunately, these essential datasets were not available for the specific area of interest, thus impeding their incorporation into the assessment. The significance of these data in providing a more complete financial perspective on fire risk is widely recognized. However, due to their unavailability within the study area, their inclusion was not feasible. Despite this constraint, this study contributes valuable insights into the broader understanding of fire risk assessment. Future research endeavors may benefit from obtaining access to these datasets, which could enhance the overall depth and scope of fire risk assessments.

In addition, updating risk layers with field data offers several advantages in terms of improving the accuracy and granularity of fire risk assessments. By incorporating real-world observations from the field, the assessment becomes more precise and context-specific, providing valuable insights into the vulnerability of specific areas to fire incidents.

As wildfires continue to pose significant challenges, the integration of advanced modeling techniques and real-world data remains essential for guiding informed decision-making and implementing effective wildfire management strategies. This study underscores the importance of ongoing research efforts to continuously improve fire risk assessments, ultimately contributing to the protection of communities and ecosystems from the devastating impacts of wildfires.

## 5. Conclusions

This research presents a contemporary and comprehensive approach to fire risk assessment at a building-block level, addressing the urgent need to combat extreme forest fire events threatening human life, the environment, and infrastructure, especially in urban and peri-urban areas. The presented fire risk assessment in the Attica Region of Greece provides an operational support tool for efficient pre- and during-fire event management, guiding stakeholders in immediate response efforts.

The methodology incorporates innovative techniques, including modeling, remote sensing (RS), geographic information systems (GIS), artificial intelligence, and machine learning (AI/ML). By integrating diverse spatial layers such as fire hazard, vulnerability, and exposure, this study offers valuable insights into the spatial distribution of fire risk in high-risk areas like Kaki Thalassa settlement.

The use of very-high-resolution data, meteorological inputs, and the parameterized FlamMap model enables a detailed analysis of fire spread patterns under various wind scenarios. The resulting worst-case fire hazard map becomes a critical tool for fire management and emergency planning, identifying high-risk areas and supporting proactive strategies to mitigate worst-case fire scenarios, enhancing community resilience and preparedness.

Despite data limitations, this study optimizes vulnerability and exposure assessment by combining national cadaster data, high-resolution RS data, and AI/ML modeling. The incorporation of field campaigns fine-tunes and validates the methodology, generating targeted management plans and pre-event suggestions.

This study's comprehensive and integrated approach surpasses existing fire management plans and guidelines in Greece, offering a building-block-level analysis. While facing challenges, such as data limitations and modeling constraints, this study's cautious approach, including educated overestimations for population safety, ensures a practical assessment.

In conclusion, this research provides valuable guidance for policy-makers, firefighting departments, and traffic police, enabling focused fire risk reduction measures and evacuation plans for high-risk areas. The methodology sets a commendable standard for future research, continually refining and expanding fire risk assessment methodologies to adapt to evolving challenges and advancements in the field.

**Author Contributions:** Conceptualization, A.Y., N.S. and C.K.; methodology, A.Y., S.G. and C.K.; software, S.G., A.Y., M.-C.T., M.K. and M.Z.; validation, N.S., A.Y., S.G. and C.K.; writing—original draft preparation, A.Y., M.Z., M.K. and N.S.; writing—review and editing, C.K. and D.H.; visualization, M.K., M.-C.T., M.Z. and A.Y.; supervision, C.K.; project administration, N.S. All authors have read and agreed to the published version of the manuscript.

**Funding:** This research work was developed under the project "Seismic, Fire & Flood Risk Assessment in Attica Region, Greece", funded by the Region of Attica (Programme Agreement, March 2021).

**Institutional Review Board Statement:** Not applicable.

**Informed Consent Statement:** Not applicable.

**Data Availability Statement:** Data and information produced as part of this project constitute intellectual property of Attica Region, Greece. Formal permission is required for data acquisition and reuse.

**Acknowledgments:** This study has been developed under the umbrella of the national research project "Seismic, Fire Flood Risk Assessment in Attica Region, Greece", led and coordinated by the Operational Unit "BEYOND Centre of Earth Observation Research and Satellite Remote Sensing" of the Institute for Astronomy, Astrophysics, Space Applications and Remote Sensing, National Observatory of Athens.

**Conflicts of Interest:** The authors declare no conflict of interest. The funders had no role in the design of the study; in the collection, analyses, or interpretation of data; in the writing of the manuscript; or in the decision to publish the results.

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
