# Peer review of "Geoinformatics and Machine Learning for Comprehensive Fire Risk Assessment and Management in Peri-Urban Environments: A Building-Block-Level Approach"

_applsci, doi:10.3390/app131810261_

Round 1

Reviewer 1 Report

This study gives an integrated approach for conducting high-detail fire risk assessment and supporting strategic planning and management of fire events, in peri-urban areas that are susceptible to forest fires. Comments can be found below,

1. How to define the so-called ‘city-block level’ in the title?

2. It would be better to highlight the major difficulties and challenges in this scenario, and the original achievements to solve them, in abstract and introduction, especially concerting the different from the previous works.

3. What are the new part of the methodology adopted in this paper, compared to the previous ones, such as EMSN041, EMSN059, FireHub, etc.?

4. More data figures should be provided rather than only the visual representation of the figures 4-8.

5. What are the limitation and uncertainty in the estimation process? What is the estimation error?

 Minor editing of English language required

Reviewer 2 Report

This research presents a contemporary and comprehensive approach for fire risk assessment at a building-block level, addressing the urgent need to combat extreme forest fire events threatening human life, the environment, and infrastructure, especially in urban and peri-urban areas. The presented fire risk assessment in the Attica Region of Greece provides an operational support tool for efficient pre and during-fire event management, guiding stakeholders in immediate response efforts.

The following remarks should be taken into consideration preparing updated version of the paper:

1. There are many spelling errors in the text, for example, constatnly in line 123 should be changed to constantly, imporvement in line 309 should be changed to improvement, inerest in line 389 should be changed to interest, etc.

2. In .2.5.3, the classification criteria for building materials are not clarified, and the specific definition of intermediate levels 2-4 should be clarified.

3. In 2.2.4, codes 18 and 22 are interpreted the same.

4. Introduction mentions that settlements in and around Kakia Thalassa are classified as very high risk, and as follows we can learn that most of the area is high risk.

5. In 2.2, in addition to the land value, should the fire risk assessment also take into account the reconstruction cost of the building and the economic loss during the reconstruction period?

There are many spelling errors in the text, for example, constatnly in line 123 should be changed to constantly, imporvement in line 309 should be changed to improvement, inerest in line 389 should be changed to interest, etc.

Author Response

Response to Reviewer 2 Comments

This research presents a contemporary and comprehensive approach for fire risk assessment at a building-block level, addressing the urgent need to combat extreme forest fire events threatening human life, the environment, and infrastructure, especially in urban and peri-urban areas. The presented fire risk assessment in the Attica Region of Greece provides an operational support tool for efficient pre and during-fire event management, guiding stakeholders in immediate response efforts.

The following remarks should be taken into consideration preparing updated version of the paper:

  1. There are many spelling errors in the text, for example, constatnly in line 123 should be changed to constantly, imporvement in line 309 should be changed to improvement, inerest in line 389 should be changed to interest, etc.

Response 1: Thank you for your helpful feedback on the spelling errors. We greatly appreciate it, and we've taken your comment into consideration to improve the quality of our work.

  1. In .2.5.3, the classification criteria for building materials are not clarified, and the specific definition of intermediate levels 2-4 should be clarified.

Response 2:  Thank you for your valuable feedback and for pointing out the need for clarification in section 2.5.3 regarding the classification criteria for building materials and the specific definitions of intermediate levels 2-4.

We have carefully reviewed your comment and have made the necessary revisions to address these concerns. In section 2.2.5.3, we have now provided a more detailed explanation of the classification criteria for building materials, and we have clarified the specific definitions of intermediate levels 2-4 to ensure a clearer understanding for readers.

  1. In 2.2.4, codes 18 and 22 are interpreted the same.

Response 3: Thank you for your insightful feedback and for bringing to our attention the similarity in the interpretation of codes 18 and 22 in section 2.2.4.

We have carefully reviewed this section, and you are absolutely correct. There was an oversight in the previous version of the document, resulting in codes 18 and 22 being interpreted identically.

To rectify this issue, we have revised the interpretation of these codes to ensure they are distinct and accurately reflect their respective meanings. This correction will improve the accuracy and clarity of our work.

  1. Introduction mentions that settlements in and around Kakia Thalassa are classified as very high risk, and as follows we can learn that most of the area is high risk.

Response 4: Thank you for bringing to our attention the misleading sentence in the Introduction section. After a careful review of your comment, we have decided to remove this sentence from the Introduction as it can potentially lead to misunderstandings and is not essential for the overall context.

Additionally, to provide further clarification regarding the sentence you mentioned, upon visual inspection of the Fire Risk Map in Figure 8, we have determined that Kakia Thalassa and its surrounding areas are categorized as very high-risk areas. In contrast, in the central area, specifically in the Keratea region, and in the western part of the map, the fire risk is classified as medium/high and low/very low, respectively.

  1. In 2.2, in addition to the land value, should the fire risk assessment also take into account the reconstruction cost of the building and the economic loss during the reconstruction period?

Response 5: Thank you very much for your comment and question regarding the exposure aspect of the fire risk assessment. We acknowledge the importance of incorporating additional datasets to gain a more comprehensive understanding of the financial implications associated with fire risk assessments.

These datasets could encompass reconstruction costs, economic losses during the reconstruction period (as you correctly pointed out), and more. Unfortunately, due to the unavailability of such data in the study area, our study could not include this valuable information.

However, in recognition of the significance of your comment, we have taken steps to address this limitation. Specifically, we have added a few sentences in lines 676-689 of the document to explicitly acknowledge the constraints related to the financial aspect of the fire risk assessment. We have also suggested plausible alternatives, such as the utilization of insurance-related information, to mitigate this limitation.

Comments on the Quality of English Language

There are many spelling errors in the text, for example, constatnly in line 123 should be changed to constantly, imporvement in line 309 should be changed to improvement, inerest in line 389 should be changed to interest, etc.

Response: Thank you for your diligent review. We have already addressed the spelling errors you pointed out in your comment. Specifically, "constatnly" in line 123 has been corrected to "constantly," "imporvement" in line 309 has been changed to "improvement," and "inerest" in line 389 has been revised to "interest," among other corrections.

Your valuable comments have played a crucial role in refining our work. With all your comments now addressed, we feel confident in the final result.  

Reviewer 3 Report

To characterize the reviewed article in brief words, it is very good paper ready to be published as it is. Its scope is extremely important for any part of the Globe suffering from forest fires. Current situation in the field of interest is adequately described on the basis of comprehensive list of references. To achieve the goals of the investigation, known analytical software and programming platforms were used but the proposed methodological framework and significant results obtained via its application represent definite step forward in fire hazard modelling and forest fire risk assessment. In my opinion, the most valuable part of the paper is a discussion on assumptions and limitations of the proposed methodology.

The article under consideration is written in very good English, no issues detected besides of just one typo (page 3, row 123 - constatnly). 

To finalize my report, I'd like to thank the authors for the big (and very important!) job done. It was great pleasure to read the article!

Author Response

Response to Reviewer 3 Comments

To characterize the reviewed article in brief words, it is very good paper ready to be published as it is. Its scope is extremely important for any part of the Globe suffering from forest fires. Current situation in the field of interest is adequately described on the basis of comprehensive list of references. To achieve the goals of the investigation, known analytical software and programming platforms were used but the proposed methodological framework and significant results obtained via its application represent definite step forward in fire hazard modelling and forest fire risk assessment. In my opinion, the most valuable part of the paper is a discussion on assumptions and limitations of the proposed methodology.

The article under consideration is written in very good English, no issues detected besides of just one typo (page 3, row 123 - constatnly). 

To finalize my report, I'd like to thank the authors for the big (and very important!) job done. It was great pleasure to read the article!

Response: Thank you very much for your thorough and positive review of our article. We greatly appreciate your feedback and your kind words about our work.

We are delighted to hear that you find the paper suitable for publication in its current form and that you consider its scope to be of utmost importance, especially in regions affected by forest fires. Your acknowledgment of our comprehensive reference list and the significance of our proposed methodology and results is truly gratifying.

We also appreciate your keen eye for detail in spotting the lone typo, which we will promptly correct.

Your appreciation for the discussion on the assumptions and limitations of our methodology is valued, as we believe that transparency in this regard is essential.

Once again, we sincerely thank you for your time and effort in reviewing our paper, and we are grateful for your positive assessment. It's been our pleasure to have your feedback, and we look forward to the possibility of seeing our work published.

Round 2

Reviewer 1 Report

The authors have replied the comments properly.

Minor editing of English language required

Reviewer 2 Report

No more questions.